# Pepsin and Trypsin Treatment Combined with Carvacrol: An Efficient Strategy to Fight *Pseudomonas aeruginosa* and *Enterococcus faecalis* Biofilms

**DOI:** 10.3390/microorganisms11010143

**Published:** 2023-01-06

**Authors:** Samah Mechmechani, Adem Gharsallaoui, Layal Karam, Khaled EL Omari, Alexandre Fadel, Monzer Hamze, Nour-Eddine Chihib

**Affiliations:** 1University of Lille, CNRS, INRAE, Centrale Lille, UMR 8207-UMET-Unité Matériaux et Transformations, 59000 Lille, France; 2Laboratoire Microbiologie Santé et Environnement (LMSE), Doctoral School of Sciences and Technology, Faculty of Public Health, Lebanese University, Tripoli 1300, Lebanon; 3University of Lyon, Université Claude Bernard Lyon 1, CNRS, LAGEPP UMR 5007, 69100 Villeurbanne, France; 4Human Nutrition Department, College of Health Sciences, QU Health, Qatar University, Doha P.O. Box 2713, Qatar; 5Quality Control Center Laboratories at the Chamber of Commerce, Industry & Agriculture of Tripoli, Tripoli 1300, Lebanon; 6University of Lille, CNRS, INRAE, Centrale Lille, Université d’Artois, FR 2638–IMEC-Institut Michel-Eugene Chevreul, 59000 Lille, France

**Keywords:** biofilm, pepsin, trypsin, carvacrol, *Pseudomonas aeruginosa*, *Enterococcus faecalis*

## Abstract

Biofilms consist of microbial communities enclosed in a self-produced extracellular matrix which is mainly responsible of biofilm virulence. Targeting this matrix could be an effective strategy to control biofilms. In this work, we examined the efficacy of two proteolytic enzymes, pepsin and trypsin, to degrade *P. aeruginosa* and *E. faecalis* biofilms and their synergistic effect when combined with carvacrol. The minimum dispersive concentrations (MDCs) and the contact times of enzymes, as well as the minimal inhibitory concentrations (MICs) and contact times of carvacrol, were determined against biofilms grown on polystyrene surfaces. For biofilms grown on stainless steel surfaces, the combined pepsin or trypsin with carvacrol treatment showed more significant reduction of both biofilms compared with carvacrol treatment alone. This reduction was more substantial after sequential treatment of both enzymes, followed by carvacrol with the greatest reduction of 4.7 log CFU mL^−1^ (*p* < 0.05) for *P. aeruginosa* biofilm and 3.3 log CFU mL^−1^ (*p* < 0.05) for *E. faecalis* biofilm. Such improved efficiency was also obvious in the epifluorescence microscopy analysis. These findings demonstrate that the combined effect of the protease-dispersing activity and the carvacrol antimicrobial activity could be a prospective approach for controlling *P. aeruginosa* and *E. faecalis* biofilms.

## 1. Introduction

Biofilms are structured microbial associations attached to surfaces and enclosed in extracellular polymeric matrix (EPS) [1]. The main property of biofilms is their high resistance to antimicrobials and other types of stress compared with the cells under planktonic state; this makes biofilms very difficult to eradicate [2]. Biofilms can be found on many types of abiotic surfaces in the medical field, causing serious problems such as chronic infections, as they can reserve and disperse several infectious agents [3,4].

Pseudomonas species are ubiquitously present in the environment, some causing infections in both plants and animals [5]. Among *Pseudomonas* species, *Pseudomonas aeruginosa* is a prevalent opportunistic human pathogen that can cause a wide range of acute and chronic life-threatening infections, particularly in patients with suppressed immune systems. *P. aeruginosa* is of specific importance because it is the main cause of mortality and morbidity for patients with cystic fibrosis and one of the main health care-associated pathogens that affect hospitalized patients, while being inherently resistant to a large array of antibiotics [6]. *P. aeruginosa* is able to produce numerous virulence factors, such as biofilm formation, whose expression is regulated by complex systems of signal transduction as a response to environmental stresses [7].

Enterococci are opportunistic pathogens frequently isolated from the normal flora of the human oral cavity, gastrointestinal tract, and the genital tract of females. Enterococci are known to readily attach to diverse medical devices and form biofilms [8]. Among the enterococcal species, *Enterococcus faecalis* is the most prevalent health care-associated pathogen and commonly causes bacteremia, urinary tract infections, infections in abscesses, peritonitis, endocarditis, and decubitus and foot ulcers. This bacterium is involved in 80–90% of all health care-associated enterococcal infections [9].

The bacterial biofilm matrix is composed of three main categories: exopolysaccharides, extracellular and associated cell surface proteins/adhesins, and extracellular DNA. It generally represents 90% or more of the dry biofilm weight, and it is involved in the attachment of cells to surfaces and the maintaining of structural integrity and hydration of biofilm. Furthermore, the resistance of biofilm-structured bacteria against antibiotics and other components of antimicrobial agents is mainly supported by the EPS matrix, which limits the transport of biocides within the biofilm [10,11]. Therefore, a novel approach for the effective inactivation of bacterial biofilm cells is required to control biofilms.

Strategies using matrix-degrading enzymes have been investigated to disrupt EPS in biofilms. Mohamed et al. [12] demonstrated that among different biological enzymes tested, proteolytic enzyme (bromelain) was the best for achieving inhibition and eradication of *Klebsiella pneumoniae* biofilms. Lequette et al. [13] evaluated papain, serine protease, α-amylase, β-glucanase, and cellulase for removing biofilms formed by different bacterial species currently present in food processing chains and demonstrated the different efficacies of these enzymes. They also confirmed that the use of enzyme combinations that target multiple EPS components improves the effectiveness of an enzymatic remover against multi-species biofilms. In addition, Fagerlund et al. [14] applied Protein K, dispersin, and trypsin to control biofilms of various staphylococcal species and confirmed the high efficiency of biofilm removal by these enzymes.

When dispersing biofilm with enzymes, the released cells and microaggregates can contaminate new areas and restart the biofilm development cycle. Therefore, enzymatic treatment should be used in combination with a killing step to treat biofilm contamination. Recently, strategies using the combination of EPS-degrading enzymes with antimicrobial agents have been investigated for potential application in biofilm treatment [15,16,17,18,19,20]. EPS-degrading enzymes can disperse bacteria embedded in biofilms for more efficient disinfection when combined with biocide agents. Targeting the matrix may also perturb the viscoelastic properties to additionally reduce biofilm cohesion and improve antimicrobial access and efficiency [21]. Such an approach using proteolytic enzymes that target matrix proteins, combined with an antimicrobial agent, has not been sufficiently investigated for effective inactivation of *P. aeruginosa* and *E. faecalis* cells in biofilms. Therefore, in this study, two proteolytic enzymes, pepsin and trypsin, capable of targeting matrix proteins, were investigated for their potential to degrade *P. aeruginosa* and *E. faecalis* biofilms and the synergistic effect when combined with carvacrol, a natural monoterpenoid phenol that exhibits a broad antimicrobial and antibiofilm activity.

## 2. Materials and Methods

### 2.1. Preparation of Bacterial Strains, Reagents, and Cell Suspensions

The microorganisms used were *P. aeruginosa* (CIP 103467) and *E. faecalis* (isolated from French cheese). The strains were stored at −80 °C in tryptic soy broth (TSB; Biokar Diagnostics, Allonne, France) supplemented with 40% (*v*/*v*) glycerol. Bacteria were pre-cultivated by inoculating 100 µL of the frozen strains of culture into 5 mL of TSB medium and incubating for 24 h at 37 °C. Then, to prepare the culture, 100 µL of the pre-culture was used to inoculate 50 mL of TSB medium and was incubated for 16 h at 37 °C with shaking at 160 rpm. After overnight culture, cells were pelleted by centrifugation (5000× *g*, 5 min, 20 °C), and then harvested cells were washed twice with 20 mL of potassium phosphate buffer (PB; 100 mM, pH 7). Finally, cells were resuspended in PB and sonicated at 37 kHz (Elmasonic S60H, Elma^®^, Singen, Germany) for 5 min at 20 °C. These suspensions were used for the preparation of bacterial suspensions of defined concentration required for each experiment. Pepsin was obtained from MP Biomedicals (Strasbourg, France). Trypsin and carvacrol (98% purity) were purchased from Sigma-Aldrich (St. Louis, MO, USA). Glycine Hydrochloric acid buffer (glycine HCL; 100 mM, pH 3) was used for the preparation of the pepsin solution, phosphate buffer (100 mM, pH 7.6) was used for the preparation of the trypsin solution, and dimethyl sulfoxide (DMSO; Sigma-Aldrich, St. Quentin Fallavier, France) at a 2% (*v*/*v*) final concentration was used for the preparation of the carvacrol solution.

### 2.2. Bacterial Susceptibility Assay

A broth microdilution method was used to determine the minimal inhibitory concentrations (MICs) of carvacrol against both bacterial strains. Bacterial suspensions were adjusted to 10^6^ CFU mL^−1^ using sterile Müller–Hinton broth (MHB; Biokar Diagnostics, Pantin, France). Firstly, 100 µL of MHB was added to each well of the microtiter plates. Then, twofold serial dilutions of carvacrol solution supplemented with DMSO were prepared in the microtiter plates over the range of 0.156 to 10 mg mL^−1^. Finally, 100 µL of the bacterial suspensions were added to each well. Bacteria were not added to the negative control, and antimicrobial was not added to the positive control containing DMSO. The plates were incubated at 37 °C in Bioscreen C with continuous agitation and OD600 nm was read every 2 h during 24 h. The MIC value was then considered as the lowest concentration that inhibits the obvious growth of bacteria in the wells after incubating.

### 2.3. Time–Kill Assessment

The time–kill assay was used to investigate the bactericidal effects of carvacrol against planktonic cells of *P. aeruginosa* and *E. faecalis*. The experiment was carried out as described by Isenberg (2004) with some modifications [22]. Briefly, *P. aeruginosa* (CIP 103467) and *E. faecalis* were grown overnight and then transferred to MHB, supplemented with carvacrol at the MIC value for each strain, in order to yield a final inoculum of 10^6^ CFU mL^−1^. A medium containing DMSO was used as a control. Then, bacteria were incubated under shaking at 37 °C for 1 min, 5 min, 15 min, 30 min, 1 h, 2 h, 3 h, 4 h, 6 h, and 24 h, respectively. At the appointed time, 100 μL were collected, serially diluted, and streaked onto Müller–Hinton agar (MHA, Difco Pont-de-Claix, France). The plates were then incubated at 37 °C for 24 h and the colony forming units (CFU) were counted. The experiments were repeated three times.

### 2.4. Assessment of the Enzymatic Effects on Biofilm on Polystyrene Surface

Biofilm formation was performed on polystyrene surfaces and quantified by a crystal violet assay, as previously described, with some modifications [15]. Briefly, 200 µL of the bacterial strain of 10^7^ CFU mL^−1^ was inoculated into a 96-well microtiter plate and incubated at 37 °C for 24 h to allow biofilm formation. Then, the wells were rinsed once by depositing and aspirating 200 μL of PB, treated with 200 µL of enzymes (final concentrations 10^−4^–10 mg mL^−1^) or 200 µL of TSB (control), and incubated at 37 °C for 30 min, 1 h, and 2 h. After incubation time, the enzyme-containing medium was discarded by pipetting, and the wells were rinsed once with PB, fixed with 96% ethanol for 15 min, and stained with 1.5% (*v*/*v*) crystal violet for 20 min. Afterwards, the unbound dye was removed by washing three times with PB, and the dye bound to the bacterial cells was solubilized in 200 μL of 33% (*v*/*v*) acetic acid. Then, 100 μL of the de-stained solution was moved to a new 96-well plate, and absorbance was measured at 595 nm using the Synergy HTX multimode microplate reader (BioTek Instruments SAS, Colmar, France). The resulting OD values were determined by subtracting the OD value of negative control (treated with TSB only) from the OD values of treated samples, and the percentage of biofilm reduction was calculated. The minimum dispersive concentration (MDC) of enzymes that is effective against both biofilm strains and the time of action required to disperse biofilms were determined. All tests were repeated three times for each strain.

The bactericidal effects of both enzymes on planktonic cells of *P. aeruginosa* (CIP 103467) and *E. faecalis* were evaluated in MHB. Accordingly, 100 µL of the bacterial strains of a final concentration of 10^6^ CFU mL^−1^ were placed in a 96-well microtiter plate containing enzymes. Then, the microtiter plates were incubated in Bioscreen C at 37 °C with continuous shaking, and the OD600 nm was measured for 24 h. The effect of enzymes on the bacterial growth was assessed by reference to the well-described turbidity. This test was repeated three times.

### 2.5. Combined Treatment for Removal of Biofilms on Stainless Steel

#### 2.5.1. Biofilm Formed on Stainless Steel Surfaces

Circular stainless steel (SS) coupons (304 L, Equinox, Paris, France) of 41 mm diameter and 1 mm thickness were used as a surface for the formation of biofilms. Prior to use, the coupons were soaked in 95% ethanol (Fluka, Sigma-Aldrich, St. Quentin Fallavier, France) overnight and then rinsed with distilled water. After rinsing, the coupons were soaked in 1% DDM ECO detergent (ANIOS, Lille, France) for 15 min at 20 °C. Then, coupons were vigorously washed with distilled water for 1 min, five times, and again three times with ultrapure water (Mil-li-Q^®^ Academic, Millipore, Molsheim, France) to totally eliminate residual detergent. Afterwards, the coupons were dried in the air before being autoclaved at 121 °C for 20 min. The sterilized coupons were placed in a sterile static biofilm system, termed the NEC biofilm system, as described previously by Abdallah et al. [23]. The formation of biofilm was initialized by placing 3 mL of bacterial suspension (10^7^ CFU mL^−1^) of *P. aeruginosa* (CIP 103467) and E. faecalis on the sterile SS coupons inside each reactor and incubated at 20 °C for 1 h in static conditions to permit bacterial cell adhesion. Afterwards, coupons were rinsed twice with PB to remove non-adhering cells. Coupons were then overlaid with 5 mL of TSB medium, and the sealed systems were incubated for 24 h at 37 °C. The old TSB medium was discarded after incubation, and the biofilm-covered coupons were washed twice with PB to remove planktonic cells. Rinsed coupons were used for quantification of the biofilm biomass, antibiofilm testing, and epifluorescence microscopy analysis. For biofilm biomass quantification, attached cells were detached in 20 mL of TS broth by using a 100 mL sterile pot. The pots were vortexed for 30 s, sonicated for 5 min at 37 kHz (Elmasonic S60H, Elma, Singen, Germany), and subsequently vortexed for 30 s. Thereafter, serial dilutions were prepared in TS broth and plated on Tryptic Soy Agar (TSA; Biokar Diagnostics, Allonne, France) plates and then incubated at 37 °C. After 24 h of incubation, the number of cells was enumerated, and the results were presented in log CFU mL^−1^. Results represent the average of three independent experiments.

#### 2.5.2. Biofilm Removal by Single and Combined Treatment

The single and combined treatments with enzymes and with an antimicrobial, respectively, were evaluated against bacterial biofilms by cell counting. For enzymatic treatment, biofilms were treated with 3 mL of pepsin and trypsin solutions with a final concentration of MDCs (1 mg mL^−1^), individually or sequentially, and incubated for 1 h at 37 °C. For antimicrobial treatment, biofilms were immersed in 3 mL of carvacrol solutions with a final concentration of ½ MICs (2.5 mg mL^−1^ for *P. aeruginosa* and 0.312 mg mL^−1^ for *E. faecalis*) and incubated at 20 °C for 1 min for *P. aeruginosa* and 5 min for *E. faecalis*. For combined enzyme and carvacrol treatment, the biofilms were first treated with enzymes, individually or sequentially, and then with carvacrol using the same concentrations and incubation times as above. The carvacrol antimicrobial action was blocked by submerging the slides in 5 mL of neutralizing solution [24]. Sessile cells were detached and enumerated as described above. Biofilms treated with glycine HCL and phosphate buffer without enzymes served as enzyme controls, and biofilm treated with DMSO served as carvacrol control. Results represent the mean of three independent experiments.

#### 2.5.3. Epifluorescence Microscopy Imaging

After the treatment of biofilms with enzymes and carvacrol (as described above), the biofilms were stained for 15 min with LIVE/DEAD BacLight kit (Invitrogen Molecular Probes, Eugene, OR, USA) in darkness following the manufacturer’s instructions. Then, the coupons were washed with distilled water, held in the dark for air drying, and observed under an epifluorescence microscope (Olympus BX43, Hamburg, Germany). The green cells were considered as viable cells and the red cells were designated as non-viable cells.

### 2.6. Scanning Electron Microscopy Analysis (SEM)

In order to investigate the effect of enzymes and carvacrol on bacterial cell morphology, the treated and untreated biofilm bacterial cells were observed using SEM (JEOL-JSM-7800FLV, Tokyo, Japan). After biofilm treatment with different compounds, cells were recovered and diluted tenfold in TS. One milliliter of the diluted cells was filtered using a 0.2 µm pore size polycarbonate membrane filter (Schleicher & Schuell, Dassel, Germany) and then fixed for 4 h at 4 °C with a cacodylate buffer 0.1 M, pH 7.0 (sodium cacodylate trihydrate (CH_3_)_2_AsO_2_Na._3_H_2_O) containing 2% glutaraldehyde. Fixed cells were dehydrated by submerging the filter in an ascending series of ethanol (50, 70, 95, and 2 × 100% (*v*/*v*) ethanol) for 10 min for each concentration and dried at the critical point. Samples were covered with a thin carbon film and observed with a microscope at 3 KV.

### 2.7. Statistical Analysis

Each test was performed a minimum of three times. Statistical significance was determined by GraphPad Prism 9.0 software by using one-way ANOVA (Tukey’s method). Values of *p* < 0.05 were considered statistically significant.

## 3. Results

### 3.1. Antimicrobial Activity of Carvacrol against Planktonic Cells of Pseudomonas aeruginosa and Enterococcus faecalis

The antimicrobial activity of carvacrol was assessed against both bacterial strains. Figure 1 shows that using DMSO at 2% (*v*/*v*) as a final concentration improved the miscibility of carvacrol in water,. The results showed that the growth of the strains was not affected in the presence of DMSO as shown in Figure 1. The MIC value of carvacrol against the *P. aeruginosa* strain was 5 mg mL^−1^ (Figure 1a). The *E. faecalis* strain was more sensitive to carvacrol with a MIC value equal to 0.625 mg mL^−1^ (Figure 1b). In a time–kill assay, a bacterial population of approximatively 8 log CFU mL^−1^ was exposed to carvacrol MIC (Figure 2). The results showed that by using the MIC values of carvacrol for each strain, no visible cells were detected after only 1 min of treatment for the *P. aeruginosa* strain (Figure 2a) and 5 min of treatment for the *E. faecalis* strain (Figure 2b).

### 3.2. Assessment of the Minimal Dispersive Concentration and Enzyme-Action-Time on Biofilm Developed on Polystyrene Microtiter Plates

Our results showed that pepsin and trypsin disassemble 24 h biofilms of *P. aeruginosa* and *E. faecalis* developed in 96-well polystyrene plates in a dose- and time-dependent manner (Figure 3). Specifically, after 1 h of treatment, pepsin and trypsin at 1 mg mL^−1^ detached 41% and 50% of *P. aeruginosa* biofilm, respectively (Figure 3a,b), and 50% and 48% of *E. faecalis* biofilm, respectively (Figure 3c,d). Results also showed that the use of an extended dispersion time (2 h) and a higher concentration (10 mg mL^−1^) of pepsin and trypsin resulted in a similar dispersion of the biofilms of both bacterial strains and that the enzymes did not completely remove the biofilms from the polystyrene surface, regardless of the treatment time and enzyme concentrations used.

In order to study the effect of pepsin and trypsin on the growth of bacterial strains, *P. aeruginosa* and *E. faecalis* were incubated in the presence of enzymes. The results showed that the growth of both bacterial strains was not affected in the presence of the enzymes at concentrations up to 1 mg mL^−1^ (*p* > 0.05) (Figure 4).

### 3.3. Quantitative Assessment of the Combined Effect of Enzymes and Carvacrol on Biofilm Developed on Stainless Steel

The single and combined antibiofilm effect of enzymes and carvacrol against the preformed biofilm of *P. aeruginosa* and *E. faecalis* strains grown on stainless steel coupons was studied using culturable count assay. Both bacterial biofilms had approximately 7 log CFU mL^−1^ of bacterial biomass. (Figure 5). After 1 h of enzymatic treatment using pepsin or trypsin at the MDC level (1 mg mL^−1^), individually or sequentially, a limited biofilm removal of both bacterial strains was shown (Figure 5). However, the *P. aeruginosa* and *E. faecalis* biofilms treated with carvacrol at ½ MICs (2.5 mg mL^−1^ and 0.312 mg mL^−1^, respectively) were significantly reduced. The *P. aeruginosa* biofilm was reduced by 2 log CFU mL^−1^ (*p* < 0.05) after the 1 min treatment (Figure 5a), and the *E. faecalis* biofilm was reduced by 1 log CFU mL^−1^ (*p* < 0.05) after the 5 min treatment (Figure 5b).

For the *P. aeruginosa* biofilm, the combined treatment using enzymes followed by carvacrol showed a substantial reduction in the culturable cells (Figure 5a). Interestingly, treatment with pepsin followed by carvacrol reduced the biofilm biomass by 3.7 log CFU mL^−1^ (*p* < 0.05). After combined treatment with trypsin and carvacrol, the reduction of the biofilm biomass was 2.8 log CFU mL^−1^ (*p* < 0.05). This reduction was more significant after sequential treatment with both enzymes followed by carvacrol treatment. Specifically, there was a significant synergistic inactivation of biofilm of 4.7 log CFU mL^−1^ (*p* < 0.05) when treated in the order trypsin, pepsin, and carvacrol (Figure 5a).

For the *E. faecalis* biofilm, the combined treatment of enzymes followed by carvacrol showed a significant inactivation with a maximum reduction of approximatively 2 log CFU mL^−1^ (*p* < 0.05) after treatment using pepsin followed by carvacrol (Figure 5b). Moreover, the sequential treatment of both enzymes, in the order pepsin–trypsin or trypsin–pepsin, followed by carvacrol treatment showed a notable and almost equal reduction in the biofilm biomass of approximatively 3.3 log CFU mL^−1^ (*p* < 0.05) (Figure 5b).

### 3.4. Qualitative Assessment of the Effect of Enzymes and Carvacrol on Biofilms Viability

*P. aeruginosa* and *E. faecalis* biofilms were stained with SYTO9 and propidium iodide (PI) and observed by epifluorescence microscopy after treatment (Figure 6 and Figure 7). Results showed that the TS-treated controls of both bacteria exhibited a thick biofilm of mostly SYTO9-stained viable cells (green bacteria) with some PI-stained dead bacteria (red bacteria). After treatment with pepsin or trypsin alone at the MDC values, the results showed a reduction in the biofilm biomass compared with the control, with a predominant SYTO9 staining (Figure 6 and Figure 7). This reduction was more significant after sequential pepsin and trypsin treatment, which showed a scattered bacterium with the presence of some non-dispersed bacterial clusters for the *P. aeruginosa* biofilm (Figure 6). Furthermore, for the *E. faecalis* biofilm, a thin layer of remaining biofilm was shown after this sequential treatment (Figure 7). Results also showed that there was no significant difference in dispersion efficiency regardless of the order of the enzymes used.

After carvacrol treatment, the results showed a significant decrease in SYTO9 staining and an increase in PI staining of the superficial layer of thick biofilm for both bacterial strains (Figure 6 and Figure 7). However, after combined treatment using enzymes followed by carvacrol, the results showed a substantial reduction in the biofilm biomass, as well as the number of viable cells. This combined treatment showed the same biofilm reduction using either pepsin or trypsin followed by carvacrol for *E. faecalis* strain (Figure 7). However, for *P. aeruginosa*, results showed that the combined treatment using pepsin followed by carvacrol induced a more substantial reduction in the biofilm biomass than the combined trypsin and carvacrol treatment (Figure 6). Furthermore, after sequential enzymatic treatment, in the order pepsin–trypsin or trypsin–pepsin, followed by carvacrol, the biomass of both biofilms and the number of viable cells were further decreased, and the remaining thin biofilms were predominantly stained by PI (Figure 6 and Figure 7).

### 3.5. Effect of Enzymes and Carvacrol on the Morphology of Biofilm Cells

The structural morphology of the biofilm cells treated with pepsin, trypsin, and carvacrol was investigated using SEM and compared with untreated cells (Figure 8). Figure 8 shows that the untreated *P. aeruginosa* and *E. faecalis* cells had normal bacilliform and coccoidal forms, respectively. Furthermore, enzyme treatment showed no alteration in their cell morphology, and the bacterial cells remained intact. However, distorted and rough surface cells were observed in both biofilms after carvacrol treatment. For *P. aeruginosa*, complete cell constriction and swelling were observed after carvacrol treatment. In addition, the *E. faecalis* cells were also shown to be injured and deformed with a hollow cell wall after this treatment.

## 4. Discussion

The EPS matrix serves as a 3D scaffold that maintains mechanical stability, cohesion, and protection against antimicrobial therapies and host effectors [21]. The use of enzymes for degrading the matrix has recently been studied for their potential applications in the control of biofilms due to their ability to disrupt the biofilm matrix and return bacteria to the more vulnerable planktonic state. The disruption of this matrix can also deplete the viscoelastic properties to further weaken biofilm cohesion and improve antimicrobial efficacy [25]. Therefore, the use of enzymes combined with biocides will enhance the biocides’ accessibility to the matrix-embedded biofilm cells, thereby reducing the dose of disinfectants needed and decreasing environmental pollution [19]. In this study, we assess pepsin and trypsin for their effectiveness in disrupting the preformed biofilms of *P. aeruginosa* and *E. faecalis*. These two proteolytic enzymes are mammalian digestive enzymes that are widely available and produced on an industrial scale for food applications.

The biofilm matrix proteins play an important role in biofilm stability and architecture in several bacterial strains [26]. Recently, proteases have been widely reported to be among the main enzymes used for their antibiofilm effect [12,14,15,27,28]. Several studies have demonstrated the high efficacy of trypsin in disrupting and controlling biofilm formation by *P. aeruginosa*, *Streptococcus mitis*, *Actinomyces radicidentis*, and *Staphylococcus epidermidis* [14,29,30,31,32]. Other research has demonstrated the high potential of pepsin to remove multi-species biofilms [33]. The proteins in the EPS matrix are considered as essential constituents of *P. aeruginosa* and *E. faecalis* biofilms, assisting in the pathogenesis and maintenance of biofilm [26,34,35,36,37].

The outer membrane proteins such as OmpA have been identified as the most abundant proteins in *P. aeruginosa* biofilm and play a crucial role in biofilm development and stability [34,38,39]. In addition, Taglialegna et al. [26] have shown that the enterococcal surface protein (Esp), a Bap orthologous protein widely produced by *E. faecalis*, contributes prominently in matrix construction and biofilm formation. Our results on the biofilm detachment grown on polystyrene microtiter suggested that the proteolytic enzymes utilized in this study interact with these major protein components of the *P. aeruginosa* and *E. faecalis* biofilms. Furthermore, results showed that these enzymes were able to destroy the biofilm after 1 h of exposure, which is in agreement with other findings that have used one hour of enzyme treatments to remove bacterial biofilms [13,15,29].

The sole and combined action of enzymes and carvacrol was studied against biofilms of *P. aeruginosa* and *E. faecalis* growing on stainless steel surfaces using cell counts and microscopic analysis. The MDC values and contact times of both enzymes were used for the enzymatic treatments. As well, the ½ MIC values and contact times of carvacrol against bacterial strains were used for antimicrobial treatments since the ½ MIC was sufficient for a combined treatment of carvacrol and enzymes, providing a good demonstration of enzyme action. The use of MIC values resulted in a reduction of the biofilm biomass in a similar manner to the combined use of enzymes and carvacrol (data not shown). Therefore, these results provide a good demonstration that enzymes were able to reduce the concentration of biocide used.

After treatment with enzymes, individually or sequentially, the viable cell count in the biofilms of *P. aeruginosa* and *E. faecalis* was not significantly affected. However, the epifluorescence analysis demonstrated that pepsin and trypsin were able to disrupt the biofilm of both bacterial strains, and this dispersion was more significant after sequential treatment using the two enzymes. The difference in the dispersal pattern between the two biofilm strains observed may be explained by the fact that *P. aeruginosa* is a bacterial strain that can exist singly. In contrast, *E. faecalis* is a bacterium that does not exist alone; it is always grouped in chains or pairs. The results also showed that the enzymes were effective in disrupting biofilms without causing cell death, as the remaining biofilms were predominantly stained by SYTO9. These microscopic observations explain why the number of viable cells was not affected after enzymatic treatments since enzymes are catalytic agents that do not exhibit antimicrobial activity and therefore destabilize biofilms without affecting cell viability. In addition, SEM analysis showed that both enzymes did not affect the morphological structure of *P. aeruginosa* and *E. faecalis* biofilm cells (Figure 8). Furthermore, our results are in agreement with the bactericidal assessment of the enzymes, which showed that the growth of bacterial strains was not affected in the presence of pepsin and trypsin. Our results are consistent with a previous study showing that the use of the proteolytic enzyme, proteinase K, for individual or sequential treatment with other types of enzymes, did not significantly affect the number of viable cells in *Escherichia coli* O157:H7 biofilm grown on a stainless steel surface, and that proteinase K had no effect on the growth rate or viability of *E. coli* cells [15]. However, Zhang et al. [40] demonstrated that the presence of trypsin hindered the growth of *Streptococcus dysgalactiae* (ATCC12388) and *Streptococcus agalactiae* (CVCC586). In addition, chymotrypsin and the trypsin/chymotrypsin complex have shown variable inhibitory effects on several bacterial strains, as they can hydrolyze the outer membrane proteins of bacteria, cause damage to the integrity of the surface structures, and result in leakage of intracellular material. Carvacrol is a monoterpene phenol abundantly found in the essential oils of many aromatic plants and is widely recognized for its strong antimicrobial properties against pathogenic and foodborne microorganisms [41,42]. The antimicrobial activity of carvacrol is mainly due to the destruction of microbial integrity, causing cell death [42,43,44]. In addition, carvacrol can interact with the DNA of microorganisms, thereby affecting the gene expression of bacteria and thus reducing their virulence factors such as toxin production and biofilm formation [45,46,47]. Our study demonstrates that after carvacrol treatment, the numbers of viable cells of both biofilm strains were significantly reduced. In addition, epifluorescence microscopy observations showed that the number of PI-stained cells (dead cells) in both biofilms was significantly increased after carvacrol treatment (Figure 6 and Figure 7). SEM electromicrographs also showed that carvacrol induced many cell surface deformations and abnormalities in the bacterial cells of both biofilms. These results clearly demonstrate the strong antimicrobial activity of carvacrol. Several previous studies have shown the high efficacy of carvacrol in controlling biofilms grown on stainless steel surfaces [48,49,50,51]. However, previous studies have shown that the use of an essential oil alone did not cause the complete disruption of *P. aeruginosa* [52] and *E. faecalis* biofilms [53]. Additionally, many studies demonstrate that the biofilm matrix principally contributes to the increased tolerance and antibiotic resistance of biofilms compared with planktonic cells [54,55,56]. Based on these studies, it is considered that the reduced disinfection efficiency of essential oils may be due to the protective barrier effect of the EPS matrix. Hence, the degradation of the EPS matrix would be a suitable strategy to improve the effectiveness of essential oils.

The results showed that the *P. aeruginosa* and *E. faecalis* biofilms were reduced more by a combined pepsin or trypsin treatment with a carvacrol treatment than by that of a carvacrol treatment alone. This reduction of both biofilms was significantly more important after sequential enzyme treatments followed by carvacrol treatment compared with treatment using a single enzyme followed by carvacrol. Such increased efficiency is also evident in the epifluorescence microscopy analysis which showed that the combined treatment using enzymes, individually or sequentially, followed by carvacrol, exhibited a synergistic effect of the enzymes’ dispersive activity and the carvacrol antimicrobial activity since biofilms were more reduced in terms of biomass and viability. It was confirmed from our results that pepsin or trypsin treatment combined with carvacrol can synergistically improve biofilm disinfection. In addition, Cui et al. [57] showed that *E. coli* biofilm was greatly reduced by essential oil in the presence of protease. Other studies have been performed using proteases and antimicrobials in tandem. They have shown that protein removal destabilizes and weakens the biofilm matrix and thus improves the susceptibility of Gram-negative and Gram-positive pathogens to antimicrobials [15,19,20,58].

Pepsin and trypsin are both mammalian digestive enzymes requiring specific amino acids in the polypeptide chain to hydrolyze proteins. Pepsin splits the polypeptide chain of at least six different amino acids between two hydrophobic preferentially aromatic residues, whereas trypsin cleavage requires the presence of specific basic amino acids, such as lysine or arginine [33]. Based on this specificity, pepsin showed more interesting results in reducing both treated biofilms when combined with carvacrol, especially for the *P. aeruginosa* biofilm (Figure 5a). This improved action was also demonstrated in epifluorescence microscopy analysis showing that the biofilm removal of *P. aeruginosa* was more significant after combined treatment using pepsin with carvacrol than with using the trypsin enzyme (Figure 6). Similarly, Marcato-Romain et al. [33] also demonstrated that pepsin was significantly more effective than trypsin in disrupting multi-species industrial biofilms. Furthermore, the difference in the efficiency of inactivation of the *P. aeruginosa* biofilm depending on the order of treatment of the two proteolytic enzymes prior to treatment with carvacrol suggests that this order is important for efficient inactivation of *P. aeruginosa* biofilm cells (Figure 5a). It might also be a reflection of the spatial or structural distribution of biofilm components.

Indeed, these two proteolytic enzymes can cause structural defects in the biofilms and degrade the barrier properties by possibly interacting with proteins, including outer membrane proteins for *P. aeruginosa* and Esp for *E. faecalis*, thereby facilitating the penetration of carvacrol and reducing the survivability of the cells. Hence, our findings suggest that proteins can be good targets for removal to allow effective penetration of disinfectants such as carvacrol to inactivate *P. aeruginosa* and *E. faecalis* cells in the biofilm.

Approaches using enzymes alone to control biofilms have drawbacks in terms of cost, high dependence of activity on environmental factors, and stability, as they can self-degrade whereby causing instability [59]. In addition, the high volatility and low water solubility of essential oils can minimize the antibacterial activity of these components [60,61]. Therefore, as an answer to these drawbacks, enzymes and carvacrol could be stabilized by encapsulation into abiotic materials. Encapsulation can enhance the antibiofilm activity of these active agents and improve their stability; this may be cost-effective if the stability achieved is sufficiently maintained for repeated use.

This work suggests that the combined treatment using enzymes and essential oil may be a promising technology for the eradication of microbial biofilm infections. The use of these natural agents would further reduce the use of chemical agents, energy costs, and water consumption needed for biofilm control.

## Figures and Tables

**Figure 1 microorganisms-11-00143-f001:**
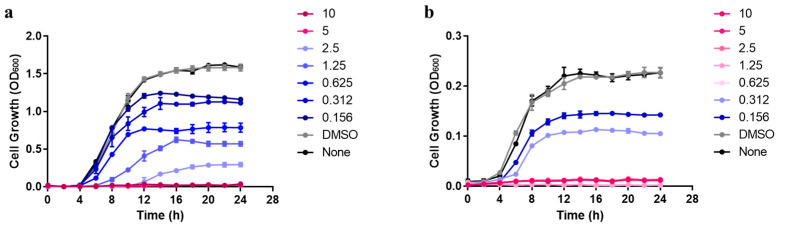
Minimal Inhibitory Concentration (MIC) of carvacrol against planktonic cells of *P. aeruginosa* (**a**) (5 mg mL^−1^) and *E. faecalis* (**b**) (0.625 mg mL^−1^). Bacterial turbidity was measured at OD600 nm using a spectrophotometer at intervals of 2 h over a 24 h incubation period with different concentrations of carvacrol in three independent experiments. Control containing DMSO was used.

**Figure 2 microorganisms-11-00143-f002:**
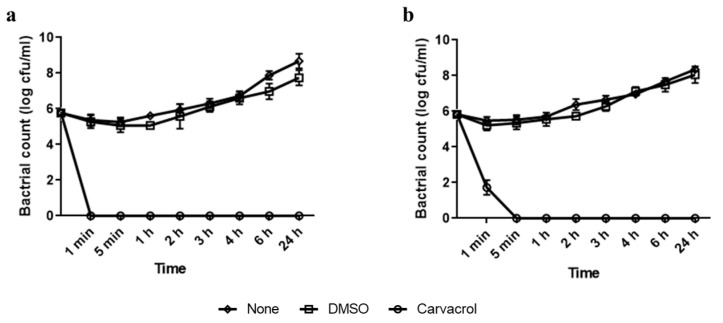
Time–kill assay curve from agar plate count method of carvacrol at MICs against planktonic cells of *P. aeruginosa* (**a**) and *E. faecalis* (**b**). Control containing DMSO was used. Results are presented as the means (±SD) of three independent experiments. “0” in the scale represents “below the detection limit”.

**Figure 3 microorganisms-11-00143-f003:**
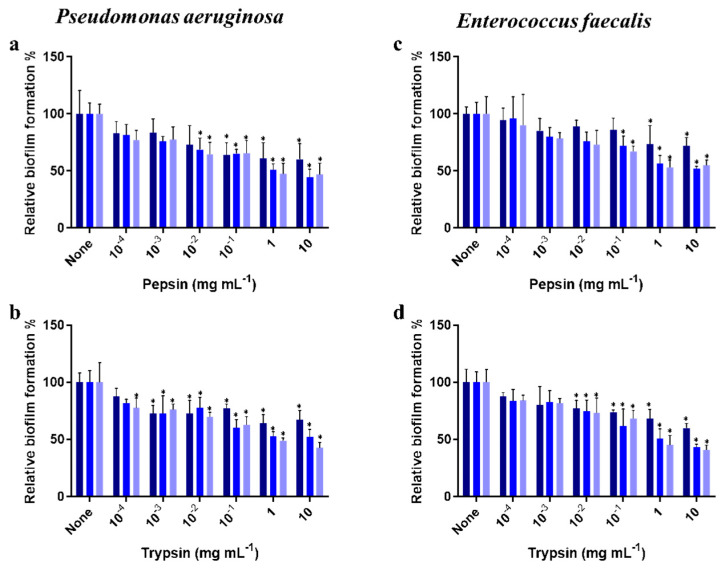
Dispersal of established *P. aeruginosa* (**a**,**b**) and *E. faecalis* (**c**,**d**) biofilms by pepsin and trypsin using different concentrations (10^−4^–10 mg mL^−1^) and different times of action (30 min−1 h−2 h). Biofilm dispersion was studied in 96-well polystyrene plates at 37 °C. Total biofilm formation was measured at OD600. Results are presented as the means (±SD) of three independent experiments. * *p* < 0.05 indicates a significant difference compared with control using Tukey’s test.

**Figure 4 microorganisms-11-00143-f004:**
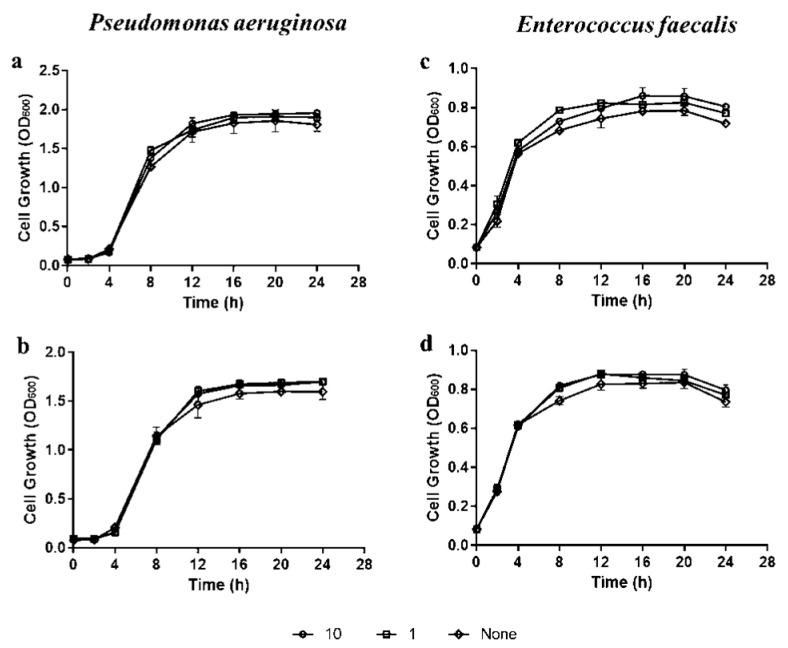
The growth of *P. aeruginosa* (**a**,**b**) and *E. faecalis* (**c**,**d**) cells in the presence of pepsin (**a**,**d**) and trypsin (**b**,**d**) cultured at 37 °C for 24 h with shaking. Bacterial turbidity was measured at OD600 nm using a spectrophotometer at intervals of 2 h over a 24 h incubation period with different concentrations of enzymes in three independent experiments.

**Figure 5 microorganisms-11-00143-f005:**
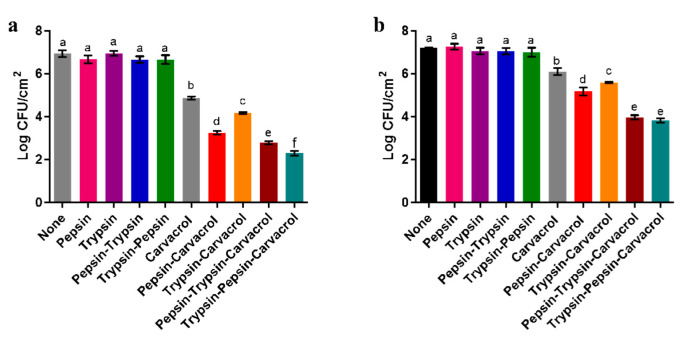
The reduction (in log CFU mL^−1^) of *P. aeruginosa* (**a**) and *E. faecalis* (**b**) biofilms after treatment with pepsin, trypsin, or sequential treatment of both at a concentration of 1 mg mL^−1^ for 1 h each, or followed by treatment with carvacrol at ½ MICs of each strain for 1 min for *P. aeruginosa* and 5 min for *E. faecalis*. Biofilms were developed on a stainless steel surface at 37 °C for 24 h. Results are presented as the means (±SD) of three independent experiments. Different letters (a–f) indicate significant differences (*p* < 0.05) using Tukey’s test.

**Figure 6 microorganisms-11-00143-f006:**
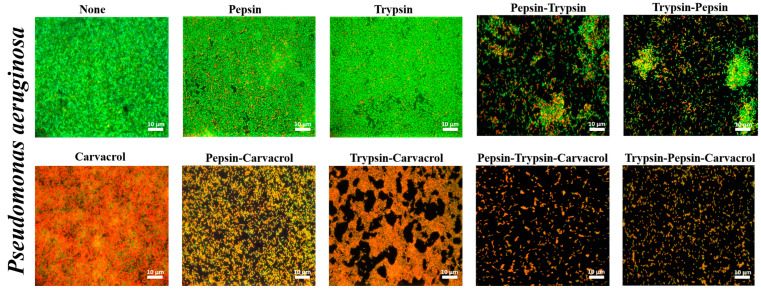
Epifluorescence microscopic images of *P. aeruginosa* biofilms after treatment with pepsin, trypsin, or sequential treatment of both at a concentration of 1 mg mL^−1^ for 1 h each, or followed by treatment with carvacrol at ½ MIC (2.5 mg mL^−1^) for 1 min. Cells were visualized after staining with SYTO9 (green fluorescence for living bacteria) and propidium iodide (red fluorescence for dead bacteria). Control represents biofilm treated with tryptone salt broth.

**Figure 7 microorganisms-11-00143-f007:**
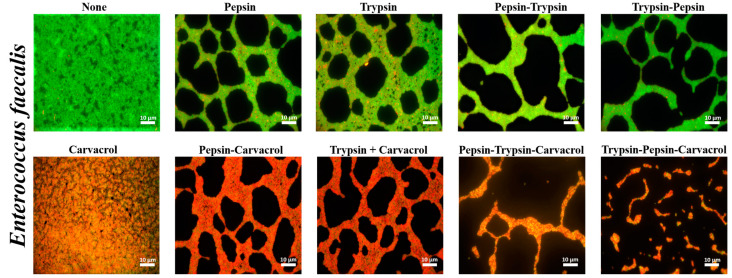
Epifluorescence microscopic images of *E. faecalis* biofilms after treatment with pepsin, trypsin, or sequential treatment of both at a concentration of 1 mg mL^−1^ for 1 h each, or followed by treatment with carvacrol at ½ MIC (0.312 mg mL^−1^) for 5 min. Cells were visualized after staining with SYTO9 (green fluorescence for living bacteria) and propidium iodide (red fluorescence for dead bacteria). Control represents biofilm treated with tryptone salt broth.

**Figure 8 microorganisms-11-00143-f008:**
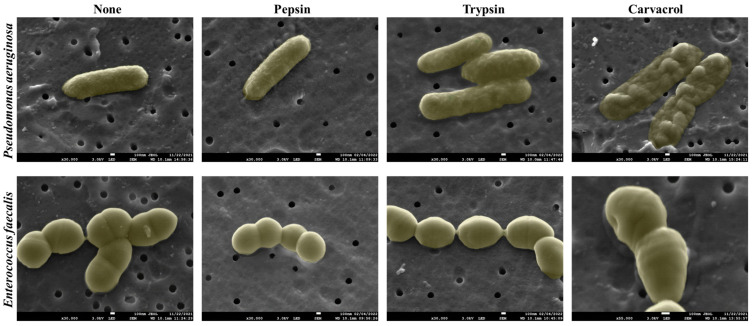
SEM micrographs of *P. aeruginosa* and *E. faecalis* biofilm cells after treatment with pepsin and trypsin at the minimal dispersive concentration (1 mg mL^−1^) and carvacrol at the MICs. Control represents biofilm cells treated with tryptone salt.

## Data Availability

The authors certify that they will comply with MDPI’s Data Policy: Data will be made publicly available upon publication and upon request for peer review.

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
