# Peer review of "Pepsin and Trypsin Treatment Combined with Carvacrol: An Efficient Strategy to Fight Pseudomonas aeruginosa and Enterococcus faecalis Biofilms"

_microorganisms, 2023, doi:10.3390/microorganisms11010143_

Round 1

Reviewer 1 Report

The study investigates the proteolytic effect of two enzymes (pepsin and trypsin) in an experimental protocol single and combined with carvacrol to induce the degradation of the polymer matrix of the bacterial biofilm produced by Pseudomonas aeruginosa and Enterococcus faecalis and to evaluate the microbicidal effect.

The introduction is focused on the proposed objectives and describes well the current state of research (state of the art) in the field.

The manuscript is well-documented and presents a precise and concise description of the experimental results and their interpretation. A significant reduction in the mass of the bacterial biofilm and viability was recorded in all situations, but much stronger after the sequential combined treatment in the order of trypsin, pepsin and carvacrol [4.7 log CFU mL-1 for P. aeruginosa and 3.3 log CFU mL-1 for E. faecalis; (p < 0.05)].

Author Response

The study investigates the proteolytic effect of two enzymes (pepsin and trypsin) in an experimental protocol single and combined with carvacrol to induce the degradation of the polymer matrix of the bacterial biofilm produced by Pseudomonas aeruginosa and Enterococcus faecalis and to evaluate the microbicidal effect.

The introduction is focused on the proposed objectives and describes well the current state of research (state of the art) in the field.

The manuscript is well-documented and presents a precise and concise description of the experimental results and their interpretation. A significant reduction in the mass of the bacterial biofilm and viability was recorded in all situations, but much stronger after the sequential combined treatment in the order of trypsin, pepsin and carvacrol [4.7 log CFU mL-1 for P. aeruginosa and 3.3 log CFU mL-1 for E. faecalis; (p < 0.05)].

The authors are thankful for the time spent by the reviewer on the revised version and warmly thank the reviewer for these positive comments

Reviewer 2 Report

The manuscript microorganisms-2097781 deals with the possibility to exploit a combination of pepsin, trypsin and carvacrol to disaggregate the biofilm from the pathogens P. aeruginosa and E. faecalis. The efficacy of the tri-combinatory was demonstrated over the use of the single (or couples) compounds and a significant reduction of the viable colonies was obtained.

The topic of the manuscript if of course of interest for the Journal’s readers as biofilm and its drug(s) resistance is a major challenge in many fields, with particular reference to the clinical field. However, some major aspects have to be clarified prior to suggest this manuscript for publication. Here my comments:

Major:

1. The Authors are asked to clarify the innovative aspects (or the improvements) of this work in comparison to their previous publication DOI:10.1080/08927014.2022.2151361 where they applied the same combination towards the same strains getting practically the same results with the only difference of the micro-encapsulation of the compounds.

2. In general, it is not clear how and for which kind of application the Authors plan/hypothesize to apply the developed combination. If a clinical application is speculated a preliminary cytocompatibility must be provided as the exploited enzymes are responsible foe ECM degradation and 1-hour exposition will be most likely cytotoxic.

3. In line with the previous comment, it is not clear the rationale of the use of stainless steel as test substrate. Please clarify this choice in relation to the final application.

4. SEM images reported in Figure 8 are definitely not representative for the biofilm 3D structure, they are random single colonies. So, new images representative for biofilm must be provided or the conclusions must be reconsidered according the real meaning of the images.

Minor:

1. Many typos are present, please check the text.

2. Please fully spell the name of the media/reagents at the first quote and check the correctness of acronyms.

Author Response

The manuscript microorganisms-2097781 deals with the possibility to exploit a combination of pepsin, trypsin and carvacrol to disaggregate the biofilm from the pathogens P. aeruginosa and E. faecalis. The efficacy of the tri-combinatory was demonstrated over the use of the single (or couples) compounds and a significant reduction of the viable colonies was obtained.

The topic of the manuscript if of course of interest for the Journal’s readers as biofilm and its drug(s) resistance is a major challenge in many fields, with particular reference to the clinical field. However, some major aspects have to be clarified prior to suggest this manuscript for publication. Here my comments:

We thank the reviewer very much for this encouraging comment on our manuscript

Major:

Point 1: The Authors are asked to clarify the innovative aspects (or the improvements) of this work in comparison to their previous publication DOI:10.1080/08927014.2022.2151361 where they applied the same combination towards the same strains getting practically the same results with the only difference of the micro-encapsulation of the compounds.

Response 1: During the PhD thesis of my student Samah MECHMECHANI the goal was to use a combination of enzyme activities and antibacterial one. First, we study the combined effect of free enzymes with carvacrol to assess the antibiofilm activity. Our objective was to study the effect of the combined dispersive activity of the enzymes and the antimicrobial activity of carvacrol in their free form this what has been submitted to your journal. In the paper published in Biofouling,  the goal was to check out whether we can improve the enzyme activities and the antimicrobial one via microencapsulation, the challenge was also to decrease the concentration of both enzyme and the carvacrol used. 

Point 2: In general, it is not clear how and for which kind of application the Authors plan/hypothesize to apply the developed combination. If a clinical application is speculated a preliminary cytocompatibility must be provided as the exploited enzymes are responsible for ECM degradation and 1-hour exposition will be most likely cytotoxic.

Response 2: The conclusive results obtained on the biofilm show that the combination of treatments is more effective than using each one separately. We chose 2 enzymes that are inexpensive, natural and widely available and carvacrol which is a natural antimicrobial agent. This combination is important in order to target the biofilm in several ways, EPS and biofilm cells. This treatment is intended to disrupt the biofilm to reduce bacteria on abiotic surfaces like the healthcare facilities, such as endoscopes, and not for humans or animals.

Point 3: In line with the previous comment, it is not clear the rationale of the use of stainless steel as test substrate. Please clarify this choice in relation to the final application.

Response 3: Stainless steel (SS) coupons will be used as a surface for biofilm formation. Stainless steel is a material to which Gram-negative and Gram-positive bacteria can adhere in a short time and that is used in many type of industrial material in the medical and food sector.

Point 4: SEM images reported in Figure 8 are definitely not representative for the biofilm 3D structure, they are random single colonies. So, new images representative for biofilm must be provided or the conclusions must be reconsidered according the real meaning of the images.

Response 4: After biofilm treatment with pepsin, trypsin, and carvacrol, the biofilm cells were diluted and filtered through a 0.2 µm pore size polycarbonate membrane filter. The purpose of this experiment is not to visualize the entire biofilm, but to visualize the effect of each treatment on the biofilm cells. That’s why, we filtered the biofilm cells in order to obtain separate and clear cells and to clearly visualize and demonstrate that the enzyme did not affect the biofilm cells while carvacrol strongly affected the shape of the biofilm cells. In our manuscript we mentioned in each conclusion biofilm cells and not biofilm.

Minor:

Point 1: Many typos are present, please check the text.

Response 1: Modified accordingly

Point 2: Please fully spell the name of the media/reagents at the first quote and check the correctness of acronyms.

Response 2: Modified accordingly

Reviewer 3 Report

Include the scale bar in Figures 6 and 7.

Mention carvacrol concentration in Figures 1a and 1b, figure caption.

The authors are required to include SEM micrographs for pepsin+carvacrol, trypsin+carvacrol, pepsin+trypsin+carvacrol and trypsin+pepsin+carvacrol treated P. aeruginosa and  E. faecalis biofilms. 

Author Response

We thank the reviewer very much for his encouraging feadback on our manuscript

Point 1: Include the scale bar in Figures 6 and 7.

Response 1: Modified accordingly

Point 2: Mention carvacrol concentration in Figures 1a and 1b, figure caption.

Response 2: Modified accordingly

Point 3: The authors are required to include SEM micrographs for pepsin+carvacrol, trypsin+carvacrol, pepsin+trypsin+carvacrol and trypsin+pepsin+carvacrol treated P. aeruginosa and  E. faecalis biofilms. 

Response 2: After biofilm treatment with pepsin, trypsin, and carvacrol, the biofilm cells were diluted and filtered through a 0.2 µm pore size polycarbonate membrane filter. The purpose of this experiment is not to visualize the entire biofilm, but to visualize the effect of each treatment on the biofilm cells. We filtered the biofilm cells in order to obtain separate and clear cells and to clearly visualize and demonstrate that the enzyme did not affect the biofilm cells while carvacrol strongly affected the shape of the biofilm cells. Our purpose here was not to demonstrate the effect of combined treatment on the biofilm. The effect of the combined treatment on biofilm was demonstrated in the epifluorescence analysis using Live dead kit.

Round 2

Reviewer 3 Report

This manuscript can be accepted in its current format.